# Reduced Etch Lag and High Aspect Ratios by Deep Reactive Ion Etching (DRIE)

**DOI:** 10.3390/mi12050542

**Published:** 2021-05-10

**Authors:** Michael S. Gerlt, Nino F. Läubli, Michel Manser, Bradley J. Nelson, Jürg Dual

**Affiliations:** 1Department of Mechanical and Process Engineering, Institute for Mechanical Systems, 8092 Zurich, Switzerland; mmanser@student.ethz.ch (M.M.); dual@imes.mavt.ethz.ch (J.D.); 2Department of Mechanical and Process Engineering, Institute of Robotics and Intelligent Systems, 8092 Zurich, Switzerland; laeublin@ethz.ch (N.F.L.); bnelson@ethz.ch (B.J.N.)

**Keywords:** fabrication, deep reactive ion etching, process optimization, reduced etch lag, high aspect ratio, small structures

## Abstract

Deep reactive ion etching (DRIE) with the Bosch process is one of the key procedures used to manufacture micron-sized structures for MEMS and microfluidic applications in silicon and, hence, of increasing importance for miniaturisation in biomedical research. While guaranteeing high aspect ratio structures and providing high design flexibility, the etching procedure suffers from reactive ion etching lag and often relies on complex oxide masks to enable deep etching. The reactive ion etching lag, leading to reduced etch depths for features exceeding an aspect ratio of 1:1, typically causes a height difference of above 10% for structures with aspect ratios ranging from 2.5:1 to 10:1, and, therefore, can significantly influence subsequent device functionality. In this work, we introduce an optimised two-step Bosch process that reduces the etch lag to below 1.5%. Furthermore, we demonstrate an improved three-step Bosch process, allowing the fabrication of structures with 6 μm width at depths up to 180 μm while maintaining their stability.

## 1. Introduction

With the rise of the semiconductor industry, cleanroom-based production technologies in silicon gained a significant increase in popularity. The ability to fabricate micron-sized structures has revolutionised numerous industries and led to previously unseen products such as MEMS [1] and microfluidic devices [2] that recently attained increasing relevance for biotechnology and biomedical research [3]. The standard fabrication steps used for the production of these microstructures consist of photolithography, etching, and post-processing such as bonding, dicing, and packaging. Given the high complexity of the etching step in current microfabrication processes, various approaches have been developed, each having their advantages and disadvantages. One of the most common processes for microstructure fabrication is deep reactive ion etching (DRIE), where physical and chemical etching are successfully combined [4]. The full potential of DRIE was revealed with the invention of a time-multiplexed alternating process of passivation and etching by Laemer and Schilp in 1996 [5]. Named after their employer, the Bosch process is nowadays one of the key processes in the silicon industry [6]. During the passivation step, a thin polymer layer, typically consisting of octafluorocyclobutan ($C_{4}F_{8}$), is deposited onto the substrate, while in the subsequent etching step, the passivation layer and the underlying silicon are etched. Due to the ions’ acceleration towards the target, significantly higher vertical etch rates can be achieved, leading to the formation of high aspect ratio trenches (Figure 1a). The characteristic scallops that result from the Bosch process are unavoidable. However, given their size in the nanometre-range, the influence of the scallops is usually negligible (Figure 1b).

One of the Bosch process’s fundamental limitations is the reactive ion etching (RIE) lag, which leads to narrow features being etched shallower than large features (RIE lag) as the gas dynamics limit the gas transport to and from structures with aspect ratios exceeding 1:1 [7]. The RIE lag, leading to differences in channel depths due to varying channel widths, is typically above 10% for features with aspect ratios in the range of 2.5:1 to 10:1 (top of Figure 2b). Recently, the RIE lag was tackled by several research groups via adding process steps or advanced timing controls, which often significantly increases process complexity and, thus, requires expensive tools [8,9,10,11]. In this work, we achieved a RIE lag reduction to below 1.5% at an etch depth of 50  μm by solely adjusting parameters of a two-step Bosch process while maintaining its simplicity and adaptability for a large variety of fabrication machines.

Another limitation of the two-step Bosch process is its low selectivity, i.e., the ratio between mask and silicon etching, which complicates the production of structures with high aspect ratios [12]. Therefore, several approaches to optimise the selectivity were developed [13,14,15]. Nevertheless, for reliable etch depths of more than 100 μm, silicon oxide masks are required, which demands an additional patterning step in a Reactive Ion Etching (RIE) machine [16,17]. To overcome this challenge, a three-step Bosch process is utilised, in which the duration of the second process step (anisotropic etching) is lowered to the time necessary to remove the polymer layer on the bottom of the trench, i.e., breakthrough step. The newly introduced third step does not rely on particle acceleration towards the target, but predominantly on the chemical etching of silicon, which reduces the etch rate of the mask and increases selectivity. Our improvement of the three-step Bosch process enables us to use a photoresist with 1.4 μm thickness to etch depths greater than 450 μm corresponding to a selectivity of more than 350. Furthermore, our results illustrate etching angles near $89.7{}^{\circ }$, which allows for the production of silicon walls with widths of 6 μm in between two microchannels to a depth of more than 180 μm while maintaining its impermeability, a factor crucial for subsequent use in microfluidics.

In this work, we present optimised two and three-step Bosch processes that tackle two important challenges of DRIE etching, i.e., the RIE lag and the selectivity. We believe that our achievements concerning the well-described and characterised processes simplify the adaptability and accessibility of state-of-the-art technologies for a broader audience.

## 2. Materials and Methods

All processes were carried out on standard single side polished silicon wafers (500±25 μm thickness, Prolog Semicor, Ukraine). To determine the limits in terms of etch depth for the Bosch-process, 2 μm
${\mathrm{SiO}}_{2}$ was thermally evaporated on the polished side of some silicon wafers.

First, the wafers were processed in a yellow room with standard photolithography processes. To enhance the adhesion of photoresist (PR), hexamethyldisilazane (HMDS) was deposited on the wafers for 30 s using nitrogen as carrier gas. Then, 4 mL of PR was dispensed on the polished side of the silicon wafers. Two different resists, namely S1813 (Shipley, England) and AZ nLOF 2070 (Microchemicals, Germany), were used. The resists were spin coated at 4000 r.p.m. for 35 s onto the wafers resulting in a resist thickness of 1.4±0.1μm and 7.2±0.2μm for the S1813 and the AZ nLOF 2070, respectively. Next, the samples were soft-backed for 1 min per micrometre resist thickness at $100\;{}^{\circ }C$ or 90 s at $120\;{}^{\circ }C$ for AZ nLOF 2070 to evaporate solvents in the PR layer. Afterwards, the wafers were re-hydrated within the cleanroom (40–50% humidity) for 10 min per micrometre.

After finishing the PR deposition, the samples were transferred to the mask aligner (MABA6, Süss Microtec, Germany) and exposed to UV-light from a mercury lamp. The contact between mask and sample was set to vacuum, to achieve the best resolution. The exposure time depends on the exposure dose of the resists and the measured intensity of the lamp. S1813 at 1.4 μm thickness required an exposure dose of 230 mJ cm and, given the UV lamps intensity of 7.4mW cm, needed to be exposed for 30 s in our machine. AZ nLOF 2070 at 7 μm thickness required an exposure dose of 170 mJ cm and, thus, needed to be exposed for 23 s in our machine. The wafers coated with AZ nLOF 2070 were heated to $100\;{}^{\circ }C$ for 90 s for the post-exposure bake. It is worth noting that the AZ nLOF 2070 is prone to produce slightly angled side walls which might induce minor chamfers during subsequent anisotropic etching.

Finally, the wafers were developed. AZ351B (Microchemicals, Germany) was diluted with a ratio of 1:5 developer to distilled water and has been utilised for the development of S1813. AZ 726 MIF (Microchemicals, Germany) has been used undiluted for the development of AZ nLOF 2070. The development was assisted by putting the Petri dish with developer solution and wafer into an ultrasonic bath (ultrasonic power set to level 3 of 10). The wafers were removed from the developer and put into a water bath to stop the development when the silicon or the silicon dioxide gets visible. This typically happened after 15 s for S1813 and after 60 s for AZ nLOF 2070. As can be seen in Appendix A, all features applied for the evaluation of the etching performance in Figure 3b were reproduced in the etch mask by the photolithography step with sufficient resolution.

For the wafers covered with ${\mathrm{SiO}}_{2}$, the samples were transferred into a Reactive Ion Etching (RIE) machine (PlasmaPro NGP80, Oxford Instruments, United Kingdom) where the complete 2 μm
${\mathrm{SiO}}_{2}$ layer of the exposed areas was removed. The utilised process consisted of two steps that were repeated twelve times. The first step (etch) consisted of 5 min ${\mathrm{CHF}}_{3}$ flow at 40 sccm, ${\mathrm{CF}}_{4}$ flow at 40 sccm, $O_{2}$ flow at 5 sccm and 130 W high frequency (HF, 13.56 MHz) power. Then, all flows and the HF power were turned of for 5 min to let the sample cool down. All process steps were carried out at $15\;{}^{\circ }C$ and at 15 mTorr.

After photolithography or RIE etching, respectively, the samples were transferred into the deep reactive ion etching (DRIE) machine (PlasmaPro Estrelas100, Oxford Instruments, United Kingdom). The processes carried out here are well described in the result section and the Appendix A of this publication. Throughout the etching procedures, the wafers have been kept at $0\;{}^{\circ }C$ using backside cooling via liquid nitrogen.

Finally, after DRIE etching, the wafers were diced into small pieces with a dicing SAW (DAD3221, Disco corporation, Germany). We used a hub-blade (FTB R46 45130, Disco corporation, Germany) and diced at 30,000 rpm with 5 mm/s to avoid damaging the small structures. However, as can be seen in Figure 2c, damage of the wafers’ surface at the top and the sides of microfluidic channels could not be avoided completely, hence the use of lower dice speeds is recommended. In case of deep trenches, it might be advisable to introduce a carrier wafer during dicing to prevent cracking the device wafer which would lead to to possible damage of the samples. Furthermore, depending on the application and to ensure subsequent device functionality, the use of a protective resist layer might be advisable.

For evaluation of the results, the diced samples were inspected under a microscope (Axioscope, Zeiss, Germany). A blue LED (505 nm) from the top and no backlight to achieve the highest possible contrast was utilised. A camera with a one-inch sensor was connected to the microscope and the sample were evaluated at two different magnifications, 20× and 40×, respectively.

## 3. Results & Discussion

### 3.1. Bosch Process Optimisation for Reduced RIE Lag

The Bosch process is one of the most common DRIE processes as it consists of only two steps while not relying on fast response times that would only be attainable with high-end equipment, therefore, making it accessible for a broader audience and a large variety of research fields. However, despite its simplistic approach, the process is capable of providing etch angles close to $90{}^{\circ }$ or even higher, enabling the production of small structures while guaranteeing their stability on the channel bottom, i.e., inside the trenches. Nevertheless, one major limitation of the Bosch process is the RIE lag where channels with different widths are not etched to the same depth due to the limited gas supply within the constricted features, an important factor that must already be considered for channels with widths smaller than 100 μm.

In this work, the RIE lag is successfully reduced following a concept initially introduced by Lai et al., which focuses on adjusting the ratio between the duration of the passivation process $t_{P}$ and etching process $t_{E}$ [10]. While the Bosch process only consists of two steps, one process cycle can be separated into three individual tasks, i.e., polymer deposition, polymer etching, and silicon etching (see Appendix A). It is essential to highlight that the polymer deposition and the silicon etching are primarily chemical and, by that, diffusion dependent, while the initial polymer etching at the bottom of the trench is of physical nature and, therefore, mainly independent of the aspect ratio.

To better understand why a substantial increase of the passivation time, in comparison to a standard Bosch process (top of Figure 2a), led to a significant reduction in RIE lag, a brief look is taken at the procedure for a wide as well as a narrow trench. In narrow structures, the polymer deposition is slower than in wide features, leading, for a given deposition duration, to a thinner passivation layer at the confined space compared to open spaces. The subsequent anisotropic physical polymer etching, however, is mostly independent of the aspect ratio. Therefore, the silicon in narrow trenches will be exposed faster than for a wide area due to the thinner passivation layer. The last step, i.e., the chemical silicon etching, is again slower in confined spaces. However, as it starts earlier than in wide structures, due to the silicon at the bottom being exposed faster, the process duration can be chosen such that, after a single cycle of the Bosch process, the wide and the narrow trenches end at the same depth. Following the assimilation of the two process durations $t_{P}$ and $t_{E}$, slight adjustments have been performed through the iterative increment of the chamber pressure during the etching step to further improve reliability and uniformity. Please refer to Appendix A for a detailed overview of all process parameters.

With these adjustments to the original process (bottom of Figure 2a), we were able to etch channels with a width ranging from 5 μm to 20 μm down to a depth of 48 μm while the smallest channel showed a difference in height of less than 1.5% compared to open spaces (Figure 2b). The roughness of the sidewalls and especially at the edges is attributed to the dicing process. The blade chipped parts of the previously undamaged channels, as can be seen in optical microscopy images (see Appendix A). In this publication, we were solely interested in the maximal etch depth and the etch angle. However, considering the importance of other quantities such as the surface roughness for subsequent applications, it might be advisable to cover or fill the trenches with protective photoresist before dicing, or to avoid dicing completely by choosing another observation and evaluation method.

Despite etch angles $\alpha >90^{\circ }$ of our modified process, the etch depth is limited. With a PR layer of 1.4 μm thickness, a maximal etch depth of 29 μm corresponding to a selectivity of 22 can be achieved. When using a ${\mathrm{SiO}}_{2}$ mask with 2 μm thickness, the maximal etch depth increased to 141 μm corresponding to a selectivity of 71 (Figure 2c). Measurements of multiple structures from different wafers revealed reproducibility of our process with etch angles of $91.1{}^{\circ }\pm 0.6{}^{\circ }$ (see Appendix A). In summary, by adjusting the etch to deposition time ratio and reducing the pressure, we reduced the RIE lag to below 1.5% and produced structures as deep as 141 μm. It is important to highlight that our approach allowed for the controlled etching of wafers with large percentages of exposed area (as high as 90%), a task which typically provides an additional challenge in MEMS fabrication.

### 3.2. High Rate Process Optimisation for Deep Etch

Due to the limitations in the etch depth of the two-step Bosch process, a three-step Bosch process for small high-aspect-ratio structures was analysed and optimised. Small structures typically rely on thin photoresist layers to ensure appropriate resolution during the preceding photolithography step as vertical resist walls are yet unobtainable, which typically limits the etch depth to around 100 μm. Therefore, the combination of deep etching with thin PR layers demands a process with high selectivity, such as achievable through the means of a three-step Bosch process. Here, the anisotropic etching step (etch 1), where ions are accelerated towards the target with platen power, solely focuses on the etching of the thin polymer layer formed on the bottom of the trench in the previous passivation step (Figure 1a). Therefore, the anisotropic etching step has a much shorter duration in comparison to the two-step Bosch process. A second etching step (etch 2) is introduced, in which the platen power, which is responsible for the ion acceleration, is switched off and the etching gas-flow (${\mathrm{SF}}_{6}$) is increased by a factor of four. In this second etching step, the etching happens predominately chemically, leading to a significant increase in selectivity. One of the main challenges of the three-step Bosch process is the fabrication of structures with etch angles below $90{}^{\circ }$, which sets a limit to the minimally producible feature size due to possible structural instabilities at the bottom of the etched features. The etch angle of a standard three-step Bosch process (Figure 3a) is around $86.0{}^{\circ }\pm 0.4{}^{\circ }$ at 100 μm etch depth (see Appendix A). Through precise adjustment of the step times, the chamber pressure, and the ICP power, we were able to increase the etch angle reproducibly by $3.6{}^{\circ }$ on average for various trench geometries (see Appendix A), rendering the etching process more vertical. The interested reader is referred to Appendix A for the incremental improvements that were performed in order to achieve the optimised process presented in Figure 3c. In a nutshell, we lowered the pressure $p_{e}$ during the second step, i.e., the breakthrough step (etch1), leading to a higher mean free path for the ions being accelerated towards the sample and, thus, decreasing the sidewall etch. Furthermore, we adjusted the passivation and etching times to our specific design, leading to an overall thicker polymer layer while still ensuring successful removal at the bottom of the microchannels. Depending on the etched design, the passivation time might require individual adjustments to prevent black silicon formation. Finally, we lowered the ICP power during the third step (etch 2) leading to a significantly improved selectivity. Please refer to Appendix A for a detailed overview of all the processes’ parameters. It is important to note that all parameters shown in Figure 3a rely on a short response time of the machine components and, therefore, high-end equipment might be necessary. With the previously introduced adjustments, we were able to achieve an etch angle of $89.6{}^{\circ }\pm 0.1{}^{\circ }$, enabling us to produce micrometer-sized pillars that maintained their stability even at 100 μm etch depth. Compared to a standard three-step Bosch process, where the smallest stable structure was a 40 × 20 ×100 μm pillar (width × length × height) (top of Figure 3b), we were able to produce a 40 × 10 × 100 μm pillar with our improved process (bottom of Figure 3b), lowering the critical feature size by a factor of 2. Please refer to Appendix A for a tilted scanning electron microscopy image of the standard process for clarification. It has to be noted, that the wafers were diced before inspection. Therefore, some structures might have been damaged due to the sample vibrations as can be seen in Appendix A. Since microfluidic chips produced for applications are subject to dicing, the presented results reflect manufacturing conditions. Furthermore, our approach demonstrates suitability for microfluidic applications. For this purpose, we etched two microfluidic channels 186 μm deep into the silicon with a distance of only 6 μm resulting in a narrow silicon wall (top of Figure 3c). To ensure the stability of the produced structures, we bonded glass to the top of the silicon wafer and applied a water flow of 7.5 mm/s through one of the microfluidic channels to prove the wall’s impermeability (see Appendix A). By increasing the distance of the two channels to 8 μm, we were able to increase the etch depth to 277 μm (bottom of Figure 3c). Finally, we were able to etch more than 450 μm deep into the silicon with a thin PR layer of 1.4 μm (see Appendix A), illustrating the high flexibility of our approach as required for complex miniaturisation in novel biomedical lab-on-a-chip devices.

## 4. Conclusions

In this work, we tackled two fundamental challenges of DRIE etching, namely the RIE lag and high aspect ratio etching of small structures, and described the optimised processes in detail. Through an improved two-step Bosch process, we were able to etch trenches of various widths within a difference in etch depth of less than 1.5% while keeping the simplicity of the Bosch process to ensure the broad and well-established applicability of this procedure. Further, we optimised a three-step Bosch process while focusing on the reproducibility of etch angles close to $90{}^{\circ }$ and improving the overall selectivity, a task crucial to allow for the fabrication of high-aspect ratio features. With our refined process, we were able to increase the etch angle in comparison to the standard process by $3.6{}^{\circ }$ on average, enabling us to etch a 40×10×100 μm pillar (width × length × height), lowering the critical feature size compared to the standard process by a factor of 2. Additionally, we etched a wall with 6 μm width in between two microchannels down to a depth of 186 μm, while maintaining impermeability to water which is of substantial importance for future lab-on-chip applications. Our novel and improved procedures are of interest to a vast number of research groups as they allow for the fabrication of novel structures and devices as well as improving the performance of already existing applications. 

## Figures and Tables

**Figure 1 micromachines-12-00542-f001:**
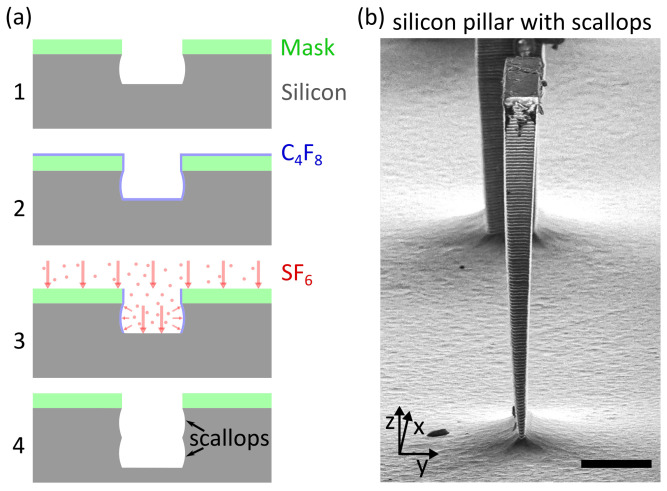
Bosch process. (**a**) Sketch of the Bosch process. (1) Silicon wafer with mask after one process cycle. (2) Deposition of a polymer layer during passivation. (3) Physical and chemical etching. (4) Silicon wafer with mask after two process cycles. (**b**) Scanning electron microscopy image of a 40 × 10 × 100 μm (x, y, z) silicon pillar. The scallops that result from the sidewall etching are clearly visible. Scale bar corresponding to 20 μm.

**Figure 2 micromachines-12-00542-f002:**
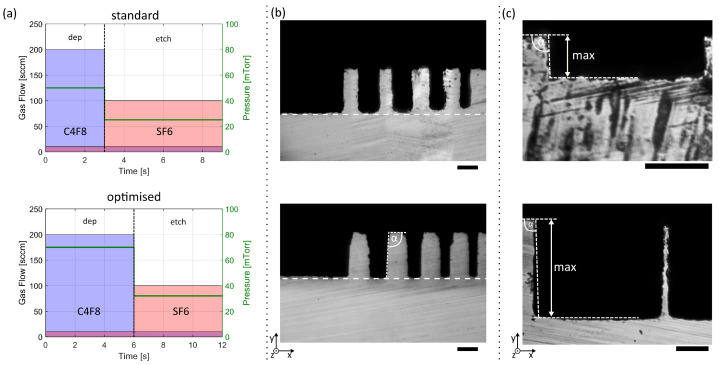
Reduced RIE lag. (**a**) Schematic of the Bosch process parameters for a standard process (**top**) and the optimised process parameters (**bottom**). A minimum gas flow of 10 sccm is always maintained to enable faster switching times (violet horizontal bar). (**b**) Optical microscopy images of diced silicon wafers to illustrate the improvement in RIE lag. (**top**) Trenches etched with the standard Bosch process, with 20 μm to 5 μm wide trenches reached a depths of 38.8 μm and 34.6 μm, respectively, corresponding to an RIE lag of 10.8%. (**bottom**) Trenches etched with our optimised process, with 20 μm to 5 μm width were etched 47.8±1.9 μm deep into the silicon wafer, corresponding to a RIE lag of below 1.5%. The optimised process lead to an etch angle of $\alpha =91.1{}^{\circ }\pm 0.6{}^{\circ }$ (see Appendix A). Scale bars corresponding to 20 μm. (**c**) Optical microscopy images of a diced silicon wafer with a 200 μm wide trench. The maximal achievable etch depth with 1.4 μm PR (**top**) and 2 μm
${\mathrm{SiO}}_{2}$ (**bottom**) as the mask was 29 μm and 141 μm, respectively. Scale bars corresponding to 50 μm.

**Figure 3 micromachines-12-00542-f003:**
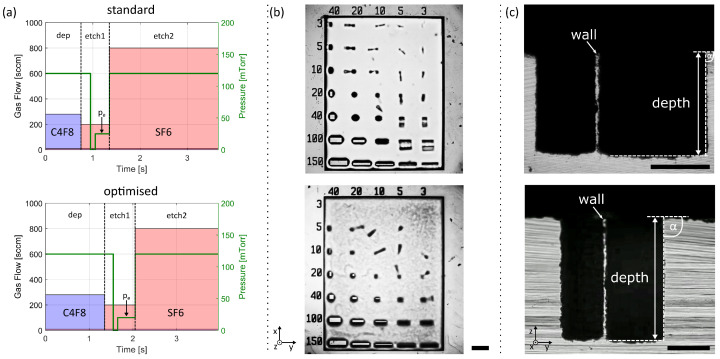
High aspect ratio etching of small structures. (**a**) Schematic of the standard three-step Bosch process parameters as provided by the supplier (**top**) and the optimised parameters (**bottom**). (**b**) Optical microscopy images of micron-sized pillars with 100 μm height (z-direction), produced with the standard (**top**) and optimised (**bottom**) process parameters. Numbers on the top and the side correspond to the width (x-direction) and length (y-direction) of the structures, respectively. With our novel process, we achieved the production of a 40 × 10 × 100 μm (x,y,z) pillar, which was impossible with the standard recipe. Scale bar corresponding to 150 μm. (**c**) Optical microscopy pictures of a diced silicon wafer with two channels and a narrow wall in-between. The tranches were etched 186 μm (**top**) and 277 μm (**bottom**) deep into the silicon wafer. The narrow wall is 6.4±0.2 μm wide (**top**) and 8.4±0.5 μm wide (**bottom**). The channels have an etch angle of $89.83^{\circ }$. The etch angle for the optimised process was $\alpha =89.6{}^{\circ }\pm 0.1{}^{\circ }$ (see Appendix A). Scale bars corresponding to 100 μm.

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
