# Peer review of "Reduced Etch Lag and High Aspect Ratios by Deep Reactive Ion Etching (DRIE)"

_micromachines, 2021, doi:10.3390/mi12050542_

Round 1

Reviewer 1 Report

The authors present an approach to reduce the RIE lag in the DRIE process by modifying the chamber pressures during the etch step. The manuscript requires some changes for further consideration. See below comments:

  1. It is recommended to include the common RIE lag information in the Abstract and the in the main manuscript. This would help readers understand the significance of the work.
  2. Ref. [5] is incomplete, please fix it.
  3. Lines 69-70: What process was used in the deposition of HMDS?
  4. Must include the process temperature of the wafer during the DRIE. Was liq. N2 used for backside cooling of the wafer in the DRIE. 
  5. The authors mention that the wafer was diced. Did they use a carrier wafer for the diced chips in the DRIE process? If so, what is the carrier wafer specs?
  6. Figs. 2 and 3: It is recommended to include images of angle measurements.
  7. It is recommended to include an image of the sample before the DRIE process.

Author Response

First, we would like to thank the reviewer for the evaluation of our revised manuscript and providing constructive feedback. Please see the attachment for a detailed response.

Reviewer 2 Report

This paper reports on an improved deep RIE process for creating high-aspect-ratio micrometer-scale structures with reduced lag.  The authors added an anisotropic etching process for removing the thin polymer layer under the increased SF6 gas flow rate with zero platen power.  The three-step etching was demonstrated to be effective for forming high-aspect-ratio pillars and walls that were stable even under water flow, thus proving its usefulness in fabricating microfluidic channels.

              The work is written nicely and can be a good contribution to the field of MEMS sensors and devices.  I will recommend publication of this manuscript after the authors consider the following points:

  1. Comparing the SEM micrographs of Fig. 2b, the trench shapes were shown to be more rough for the new etching process. Although the new process may give reduced etching lag, it is not useful if the resulting morphologies of the fabricated structures are so irregular in shape.  The authors should clearly explain if there are any tradeoffs in changing the process parameters.
  2. Why the optical images in Fig. 3b were taken at different angles? To compare the standard and optimal processes, they should use images taken from the same angle at the same magnification.
  3. Although Figs. 3c top and 3c are nice, one example is not enough to evaluate the effectiveness of the procedure. They should estimate the fabrication error and yield for the standard and new recipes.
  4. How the other parameters such as gas pressure and power affect the quality of deep RIE? The authors should show the results and discuss their effects in a quantitative manner.
  5. On page 4, “sscm” should be “sccm”.

Author Response

(The authors gave the same response as above.)

Reviewer 3 Report

The manuscript entitled “Reduced Etch Lag and High Aspect Ratios by Deep Reactive Ion Etching (DRIE)” from Gerlt et al. describes a modified three-step Bosch process for high aspect ratio etching of silicon. The paper has original results that are interesting and have applications in a broad range of research fields as well as industrial applications. I suggest publication after the following minor points are addressed.

  • I believe when the authors wrote “6 µm thickness at depths up to 180 µm …” on line 8 they meant “6 µm wide …” I would use width to describe lateral dimensions instead of thickness, which indicates out of plane dimension normally and is confusing when used together with depth. Please correct throughout the manuscript. In fact, in other places in the MS (such as line 126) authors seem to use “width” for the same dimensional property. In any case, it is better to be consistent.
  • The authors used a combination of active and passive language, which is very atypical of scientific literature. I would like them to stick to one, preferably the passive language. I encourage them to use the active language only once or twice in the MS to really distinguish what was novel in this work.
  • From Figure 2, it appears that the silicon feature edges are very rough. Is that a result of the proposed etching process? or were chipped by the dicing process? If it is the latter, for the future, I suggest embedding the grating in a polymer prior to dicing. Also, polishing after dicing the embedded structures will form clean cross-sectional images.
  • I would be interested in reading about how the optimization parameters were selected. What was the DOE strategy? Did you use some heuristic approach or was there a more strategic and systematic approach to the optimization process? Also, is it possible to collect all the parameters that were tried out along with the type and thickness of the mask, smallest feature size, maximum aspect ratio, side-wall angle, roughness and etch depth and rate (and other relevant info), in a table form? It could be in the supporting info too but it has to be there in my opinion. It is quite difficult to track down all the process parameters from the text and figures. (I didn’t have access to the SI btw)
  • There seems to be more than one conclusion of the paper. So the section title 4 should read Conclusions, not conclusion.

Author Response

(The authors gave the same response as above.)

Round 2

Reviewer 1 Report

Thanks for making the suggested changes.

Reviewer 2 Report

The authors have revised the manuscript following the comments raised.  I now ercommend publication of this paper.